# 25 years IPCC Data Distribution Centre at DKRZ and the Reference Data Archive for CMIP data

Martina Stockhause[1], Michael Lautenschlager[1]

[1]German Climate Computing Center (DKRZ), Hamburg, 20146, Germany

*Correspondence to*: Martina Stockhause (stockhause@dkrz.de)

**Abstract.** The Data Distribution Centre (DDC) of the Intergovernmental Panel on Climate Change (IPCC) celebrates its 25th anniversary in 2022. DKRZ is the only remaining DDC Partner from the original group jointly managing the DDC. In spite of changes in prioritization, it has been supporting the IPCC Assessments and long-term preserving the quality-assured, citable climate model data underpinning the Assessment Reports over these years. An active and engaged collaborative community
achieved advances in data standardization, data management best practices, and infrastructure developments. These evolving standards are reflected in the activities of the DDC. The introduction of the IPCC FAIR Guidelines into the current Sixth IPCC Assessment Report (AR6) has significantly changed the role of the DDC Partner DKRZ from an independent partner for long-term data preservation into an active partner involved in IPCC's Sixth Assessment cycle. As a result, the DDC has gained exposure and visibility, posing a challenge and an opportunity to operationalize IPCC's FAIR Guidelines and long-term
preservation approaches. While the value of DDC services has been recognized, DDC sustainability remains unresolved and is currently being discussed within the IPCC as part of a general AR6 review process to formulate recommendations for the AR7 data management.

## 1 History of TG-Data and the IPCC DDC

The current Data Distribution Centre (DDC, https://ipcc-data.org/, last access: 29 June 2022) of the Intergovernmental Panel
on Climate Change (IPCC) is jointly managed under a Memorandum of Understanding (Xing, 2021) by four partners: the German Climate Computing Center (DKRZ, Germany), the Center for International Earth Science Information Network (CIESIN, USA), the Spanish Research Council (CSIC, Spain), and MetadataWorks (UK). DKRZ is the only remaining founding partner, and it has been now operating the DDC for 25 years. The DDC is overseen by the Task Group on Data Support for Climate Change Assessments (TG-Data, https://ipcc.ch/data, last access: 29 June 2022), which is a non-permanent
part of IPCC's structure (Fig. 1). TG-Data member experts are complemented by ex-officio Members representing the DDC Partners and the Technical Support Units (TSU) of the three IPCC Working Groups (WG). The former UK DDC Partner Centre for Environmental Data Analysis (CEDA) continues its contribution to the AR6 with financial support from the WGI TSU. The core role of the DDC is the support for IPCC authors and users of data and scenarios underpinning IPCC outputs (see DDC Guidance, IPCC, 2018a). DDC makes special efforts to support users in developing countries.

The DDC was formally established at the Thirteenth Session of the IPCC (IPCC-13) on 22 and 25-28 September 1997 in the Maldives (IPCC, 1997). Deutsches Klimarechenzentrum (DKRZ) in Germany and the Climatic Research Unit (CRU) in the United Kingdom were selected to execute shared DDC operation and the Finnish Meteorological Institute (FMI) was to contribute guidance and training (see Fig. 2). During the Second Assessment Report (SAR; IPCC, 1995) cycle, IPCC Working Group II (WGII) had requested for lowering the barriers to using data from future climate scenarios provided by the Coupled
Model Intercomparison Project (CMIP) of the World Climate Research Programme (WCRP). On the IPCC Workshop on Regional Climate Change Projections for Impact Assessment (London 24-26 September 1996) and subsequent meetings of the established IPCC Task Group on Climate Scenarios for Impact Assessment (TGCIA), requirements for data availability,

data standardization and data quality together with the need for guidance materials were formulated. TGCIA recommended the establishment of a DDC to the IPCC Bureau in the same year. The IPCC Bureau requested governments on 22 July 1997

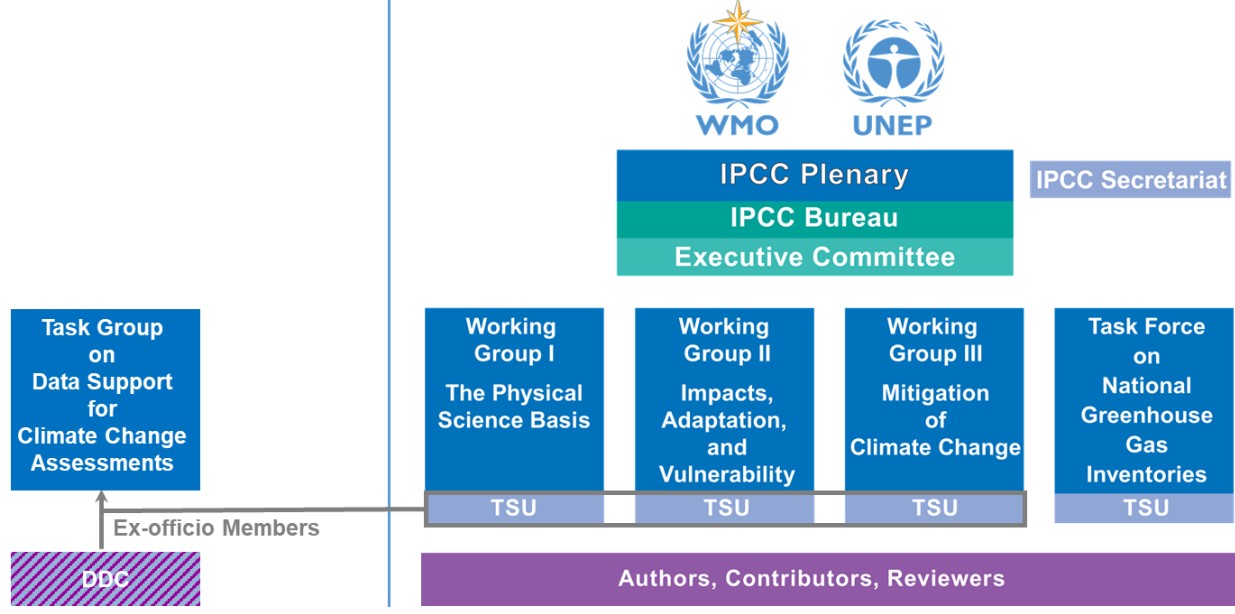


**Figure 1: IPCC Structure (from IPCC webpage)**

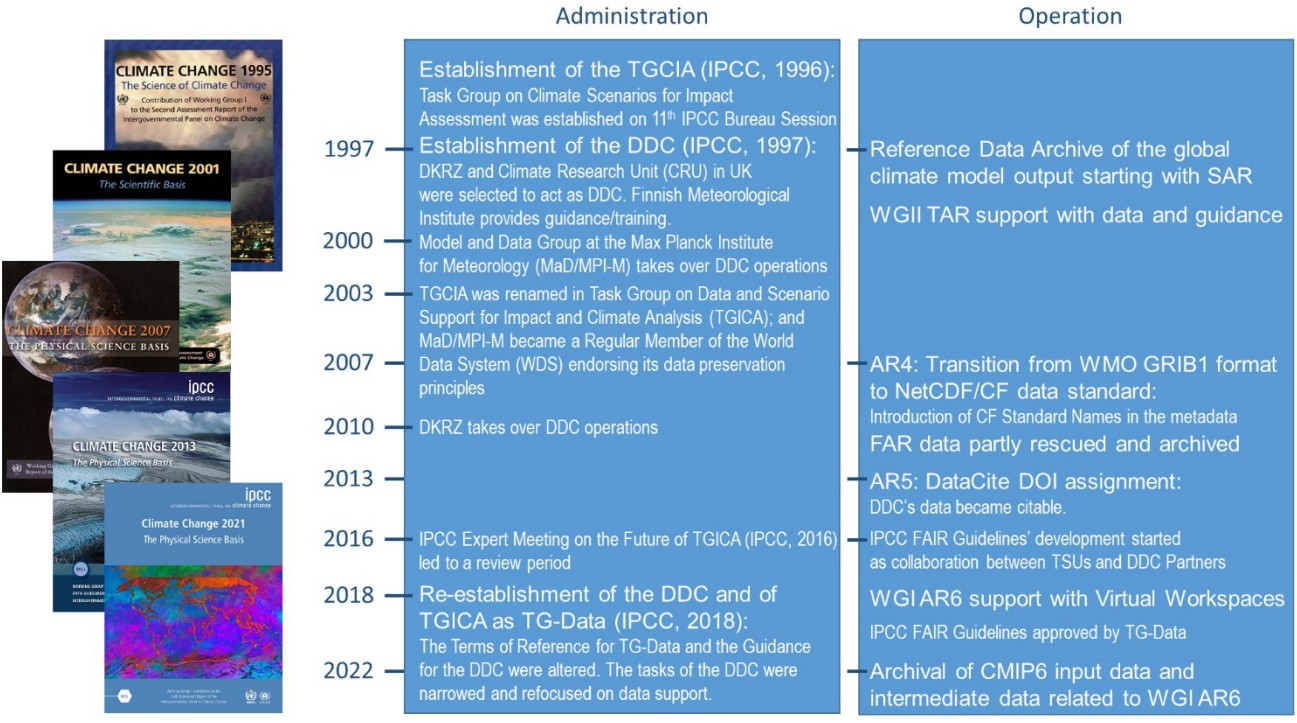

**Figure 2: IPCC DDC history and main achievements of the DDC Partner DKRZ over the past 25 years (images from cover pages of the IPCC ARs starting with SAR 1995)**

to nominate institutions to act as the DDC and to provide the necessary financial support to establish and maintain the DDC function.

The DDC Partners agreed that DKRZ should be responsible for the global climate model data related to CMIP, while the other DDC Partners took up the responsibility for other datasets in support of the IPCC WGs. Atmospheric near surface variables were collected, aggregated and disseminated by the DDC together with guidance material for the Third Assessment Report

(TAR; IPCC, 2001). After this successful operation of the DDC, the IPCC Bureau received data requests from WGIII and WGI. WGIII's data request was similar to that of WGII and could be integrated into the existing DDC service while WGI's

data needs were more complex, requesting most of the CMIP datasets. The Program for Climate Model Diagnosis & Intercomparison (PCMDI) accepted to establish the CMIP3 data archive in support of WGI. In cooperation with PCMDI, the DDC Partner DKRZ extracted the CMIP3 data subset for WGII and WGIII from the CMIP3 archive for the Reference Data Archive of the Fourth Assessment Report (AR4; IPCC, 2007). In the Fifth Assessment Report (AR5; IPCC, 2013) cycle, increasing CMIP5 data volumes led to the development of a federated data archive and the ESGF (Earth System Grid Federation) data infrastructure. The European contribution was coordinated by IS-ENES (Infrastructure for the European Network for Earth System modelling). The DDC refocused on the long-term preservation of the CMIP5 data underpinning the AR5 by transferring the CMIP5 data at the WGI snapshot date into the DDC AR5 Reference Data Archive. For nearly two decades, the IPCC DDC has closely cooperated with the CMIP data infrastructure.

The Task Group TGCIA was renamed the Task Group on Data and Scenario Support for Impact and Climate Analysis (TGICA) in 2003. TGICA developed a vision paper in 2015 and described the challenges TGICA was facing related to its limited capacity to deliver on its mandate. The IPCC Panel decided to revise TGICA's mandate and to hold an IPCC Expert Meeting on the Future of TGICA (IPCC, 2015, 2016). Vaughan (2016) summarizes TGICA's challenges and emphasizes that TGICA needs to be strengthened to be able to contribute in new ways to improve the access and use of climate data and scenarios for research and decision making through the DDC. A sharpened mandate, the clear identification of specific goals, and a realistic sense of the resources required to accomplish these goals are recommended. An IPCC Ad-hoc Task Force TGICA took up the results from the IPCC Expert Meeting and formulated revised Terms of Reference for the reestablished Task Group on Data Support for Climate Change Assessments (TG-Data) and a revised Guidance for the DDC, which were approved on IPCC-47 in March 2018 (IPCC, 2018b). The focus of the DDC was narrowed to data support tasks.

## 2      The Reference Data Archive at the DDC at DKRZ

DKRZ is the DDC Partner responsible for the long-term preservation of the global climate model data provided by CMIP. Starting with the Second Assessment Report (SAR; IPCC, 1995), core variables for the characterization of the state of the Earth System (table A1 in appendix A) from model projection of the future climate were long-term archived at DKRZ, building the Reference Data Archives for the global climate model data underpinning IPCC's ARs.

During data archival, the data are stored on tape and the metadata are enriched and quality-assured to provide sufficient and high-quality information for various downstream users without specific knowledge of climate model applications. Added metadata include context information on project, experiments and models as well as discovery information on spatial-temporal coverage, parameters and contact information. In SAR, this information was gathered mostly from the data providers by the DDC. With the increasing level of organization and standardization of CMIP, this labor-intensive and non-standardized metadata gathering from data providers could be partially replaced by machine-access of CMIP resources, e.g. accessing the ESGF index.

The size of the Reference Data Archives for the different ARs increased from around 10 GBytes and 400 datasets for SAR and TAR to ca. 1 TByte and 1 500 datasets for AR4 and then to 1.7 PBytes and 910 000 datasets for AR5 (Fig. 3). The reasons are an increased number of archived variables per model run, an increased number of models participating in CMIP, and the inclusion of daily and sub-daily data in addition to monthly data. In collaboration with NCAR (National Center for Atmospheric Research), a subset of data underpinning the FAR was rescued from NCAR's data archive in the original formats and added to the DDC in 2008. Because of the low level of standardization, these datasets are difficult to (re-)use. Data underpinning the IPCC Special Report on Global Warming of 1.5°C (SR1.5; IPCC, 2018c) were transferred into the DDC Reference Archive in 2018. The archival of the CMIP6 data subset underpinning the AR6 is ongoing. Download statistics show the long-term interest of users in the DDC Reference Data (Fig. 4).

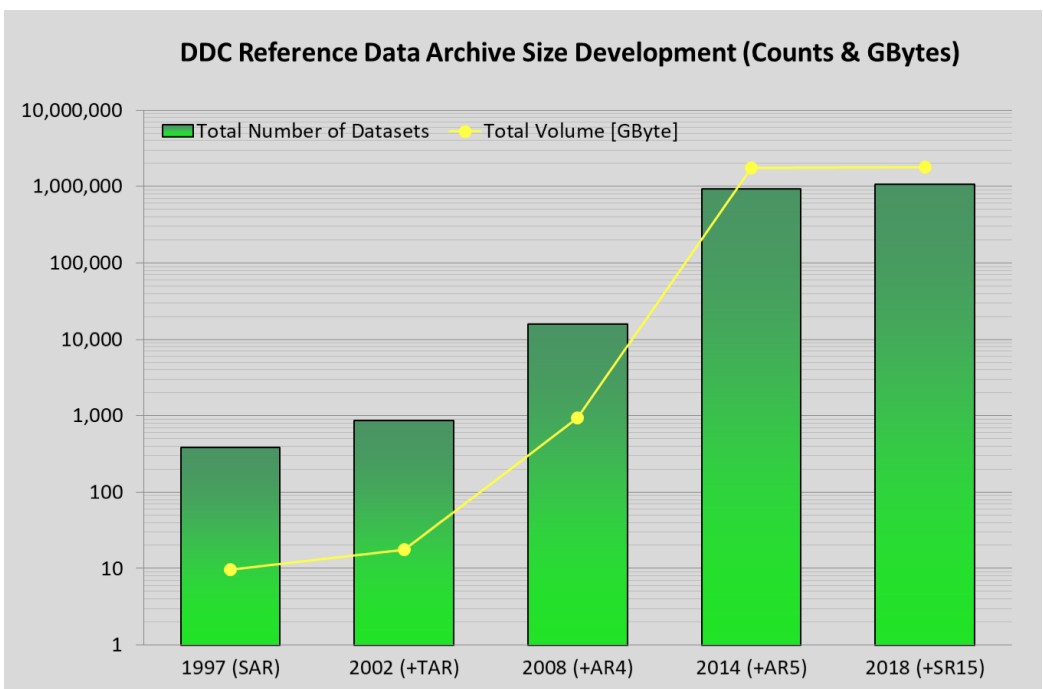

**Figure 3: Size development of the DDC Reference Data Archive for the global climate model data.**

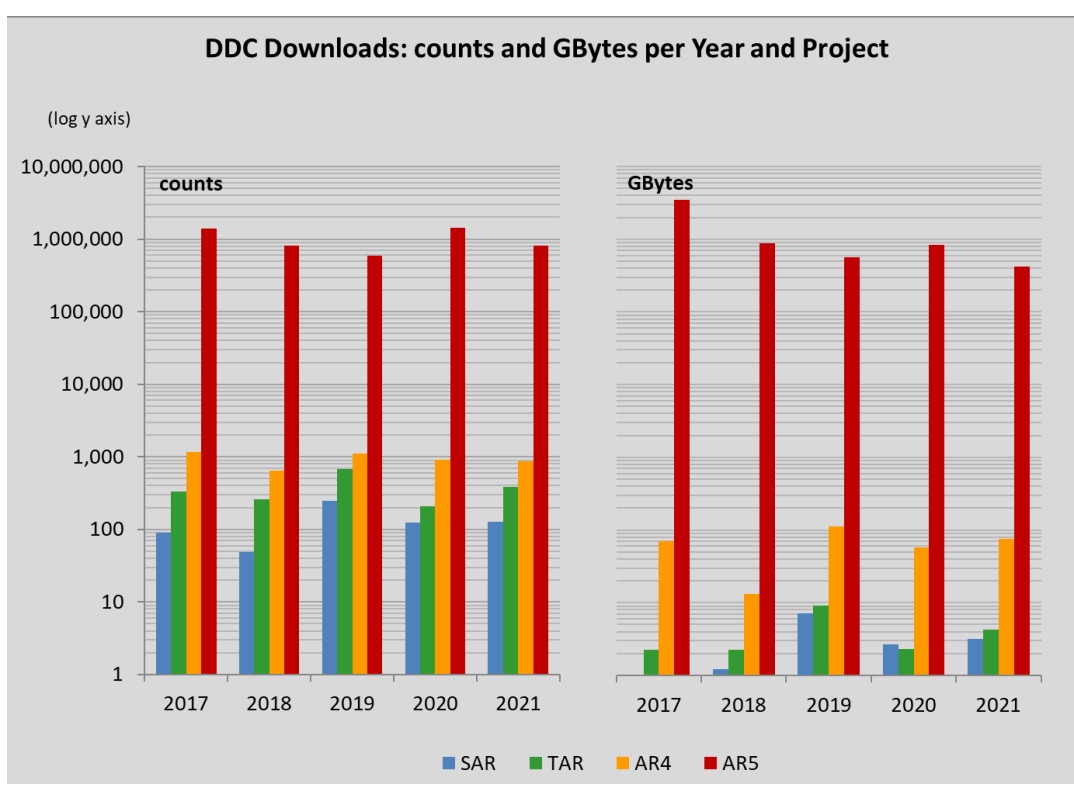

**Figure 4: Downloads in number of datasets [counts] and data volume [GByte] from the DDC Reference Data Archive over the last 5 years per Assessment Report (FAR is left out because of the incomplete Reference Archive and AR6 including SR1.5 because of the ongoing data archival; Stockhause, 2022).**

As a DDC Partner, DKRZ has committed to ensuring its DDC data remains accessible and reusable over the long-term, which involves cyclic renewal of hardware, continuous maintenance of software, and metadata and data curation. A copy of the DDC data is stored off-site at the Max Planck Computing and Data Facility (MPCDF) in Garching, Germany. New generations of hardware (tape system), for example, require the copying of the DDC data holdings on new cartridges. Software updates for data discovery, access and exchange are required to comply with new standards and interfaces in order to enhance the user experience and to meet evolving user needs. An example for a metadata curation measure was the addition of Climate and Forecast (CF) standard names to the metadata of the DDC SAR and TAR Reference Data Archives. To overcome the data

volume barrier for DDC data reuse of IPCC users located in developing countries with low internet bandwidths, the DDC introduced a service whereby users can order a set of preselected variables for seven regions on DVD and USB devices.

DKRZ adjusted to evolving best practices for data management. The DKRZ long-term data archive including the IPCC DDC Reference Data Archive was approved in 2003 as WDC Climate (WDCC) by the ICSU World Data Center system. DKRZ became a Regular Member of the ISC World Data System (WDS, https://www.worlddatasystem.org/, last access: 29 June

2022) in 2008, the year of WDS's establishment. Therefore, the DDC Partner DKRZ complies with WDS's common research repository standards. With the founding of DataCite in 2009, registering data DOIs in order to make data citable became a community expectation, which was taken up by the DDC Partner DKRZ for the AR5 Reference Data Archive published in 2013 and 2014. The long-term archival of AR5 provided further major changes in the workflow due to the extremely high data volume and several changes in the CMIP5 data infrastructure

(https://pcmdi.llnl.gov/mips/cmip5/, last access: 29 June 2022; Taylor et al., 2012):

- The data were disseminated by the newly developed federated and decentral infrastructure of the Earth System Grid Federation (ESGF, https://esgf.llnl.gov, last access: 29 June 2022; Williams et al., 2016);

- detailed model and experiment documentations were gathered from the CMIP5 participants by the Earth System Documentation project (ES-DOC, https://es-doc.org, last access: 29 June 2022: Lawrence et al., 2012); and

120        - a three level quality control procedure (CMIP5 QC, https://cmip5qc.wdc-climate.de, last access: 29 June 2022) was applied to ensure basic data quality, the consistency of metadata, and metadata conformance with community standards like NetCDF/CF and project standards like the Data Reference Syntax (DRS). Passing the three quality control levels was the prerequisite for the acceptance by the IPCC DDC for the IPCC AR5 Data Reference Archive (Stockhause et al., 2012).

The size of the CMIP5 data archive required a high level of automation for metadata and data ingest as well as for the quality control checks. New interfaces to the infrastructure components ESGF, ES-DOC and DataCite had to be developed for insertion of use and discovery metadata and data DOI registration. The long-term archived DDC AR5 data were made searchable and accessible through the ESGF, which has become the standard infrastructure for climate-related data. ETH Zurich collected a CMIP5 data subset in support of the IPCC AR5 authors in an alternate data structure. Due to difficulties

relating the individual datasets back to the CMIP5 reference datasets, the DDC AR5 Reference Data Archive was supplemented by an IPCC Working Group I AR5 snapshot. Discussions with the ETH Zurich provided valuable input for the IPCC FAIR Guidelines adopted for AR6 and the long-term archival of the CMIP6 input data in the DDC.

The DDC relies in its efforts and services on data provided by CMIP6 participants and on the standardization community efforts of several organizations and institutions. PCMDI led the AMIP and CMIP data standardization, other groups worked

on the NetCDF/CF data standard (https://cfconventions.org, last access: 29 June 2022), the CoreTrustSeal research repository standard (https://www.coretrustseal.org/, last access: 29 June 2022) or the DataCite DOI data publishing standard.

## 3       AR6 and the IPCC FAIR Guidelines

TGICA was under review of the IPCC from the start of the Sixth Assessment Cycle in January 2016 until the re-established TG-Data held its First Meeting in November 2019. This was little less than a year prior to the original WGI literature and data

cut-off date of 30 September 2020, which was postponed to 31 January 2021 due to the COVID-19 pandemic. The lack of the coordinating task group hampered the formulation and implementation of the FAIR Guidelines for the Sixth Assessment Report (AR6).

The idea for adopting the FAIR Guidelines was born during the IPCC Expert Meeting on the Future of TGICA in January 2016. The aim was to enhance the transparency of the IPCC AR6 and thereby contribute to IPCC's integrity. The IPCC FAIR

Guidelines implement the established data management principles of FAIR (Findable, Accessible, Interoperable, Reusable;

Wilkinson et al., 2016) for data and TRUST (Transparency, Responsibility, User Focus, Sustainability, Technology; Lin et al., 2020) for repository operations into the Sixth Assessment cycle. The FAIR data principles describe requirements for datasets to become an integral part of the research environment. The TRUST principles for repositories and its implementation in the CoreTrustSeal complement these essential data properties by best practices for repository operations in long-term data preservation and data stewardship.

The development of the FAIR Guidelines started at the First IPCC AR6 Data Workshop in Hamburg, Germany, 19-20 September 2017 (Stockhause et al., 2017) and continued at the second virtual meeting on 20 February 2018. In collaboration with the WDS, which started at the Data Repository Day 2018 (WDS, 2018), the FAIR Guidelines concept was formulated in Stockhause et al. (2019). This concept was discussed with IPCC authors of WGI and WGII on the IPCC Expert Meeting on Assessing Climate Information for Regions in Trieste, 16-18 May 2018 (IPCC, 2018d). The implementation of the FAIR Guidelines into tools supporting the authors was the topic of a WGI Training on Data and Software Development in Oberpfaffenhofen, Germany, 6-7 June 2019 (IPCC, 2019). An early draft of the FAIR Guidelines were formally approved by TG-Data on its first meeting in Montreal, Canada, 6-8 November 2019, and the official version 1.0 was adopted by TG-Data in a virtual meeting in 2022.

The IPCC FAIR Guidelines (Pirani et al., 2022) call for increased attention to three aspects:

• Traceability of key statements and of figure and table creation: Information on input datasets like CMIP6 (Eyring et al., 2016), final data displayed in figures, and analysis scripts generating the figures are collected from the authors by the WGI AR6 TSU. This information is recorded for every figure as part of the Supplementary Materials associated with each chapter. Moreover, bidirectional references between the digital AR6, final datasets and input datasets will enable users to navigate between these AR6 products (Fig. 5).

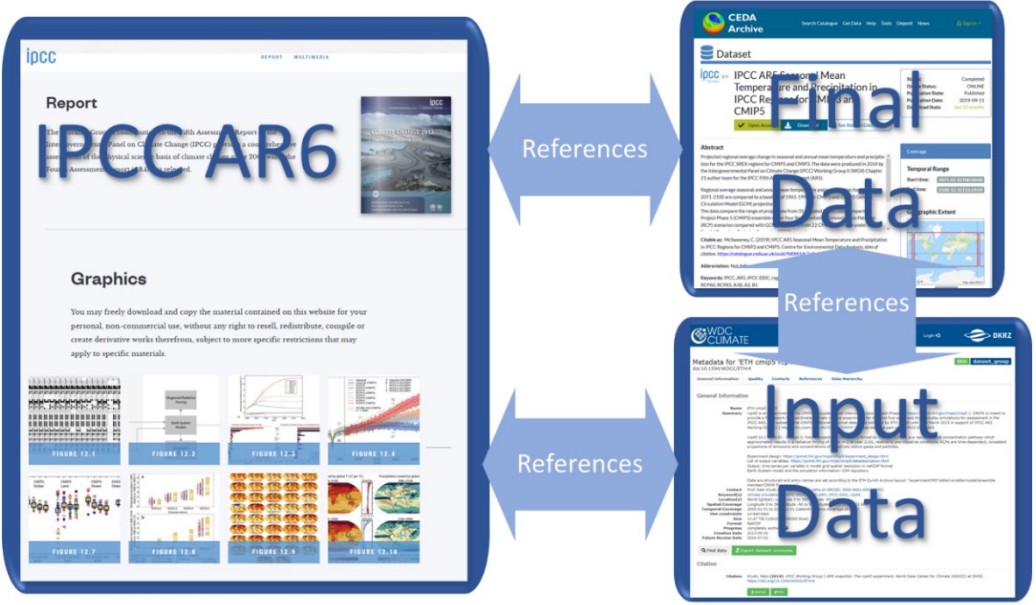

**Figure 5: Schematic vision of the bi-directional references between report, input data and final datasets in IPCC AR6 enabling users to navigate among these AR6 results (screenshots from IPCC, CEDA and DKRZ webpages).**

• Credit for input data: Input datasets used by the authors are cited in the AR6 in compliance with Good Scientific Practices (DFG, 2019). In case of CMIP6 data, data citation is required by the Creative Commons licenses (CC, https://creativecommons.org/, last access: 29 June 2022), under which CMIP6 data were published. CMIP6 data are cited in a summarized form in Appendix II of the WGI AR6 (https://www.ipcc.ch/report/sixth-assessment-report-working-group-i/, last access: 29 June 2022; IPCC, 2021), the provenance metadata of the IPCC WGI Interactive Atlas (https://interactive-atlas.ipcc.ch/, last access: 29 June 2022), and for each figure in the Supplementary Materials.

•    Long-term preservation of input data, scripts, and final data: The information, scripts and final datasets collected by the WGI TSU are transferred to the designated repository for long-term preservation. DOI registration makes the data and scripts citable and enables data users to give credit to chapter scientists for them. In case of CMIP6, the TSU compiled dataset lists for the DDC Partner DKRZ based on the data usage information collected from the authors. For data long-term archival, the listed CMIP6 datasets are replicated, use metadata are accessed from the ESGF, and further documentations from the
Citation Service (Stockhause and Lautenschlager, 2017) and if available from ES-DOC (Pascoe et al., 2020). The long-term archival workflow is depicted in Figure 6.

The implementation of the IPCC FAIR Guidelines required a close cooperation between WGI TSU and the DDC Partners and relied on the CMIP6 infrastructure partners and information provided by the CMIP6 participants as well as on the information compiled by the IPCC authors.


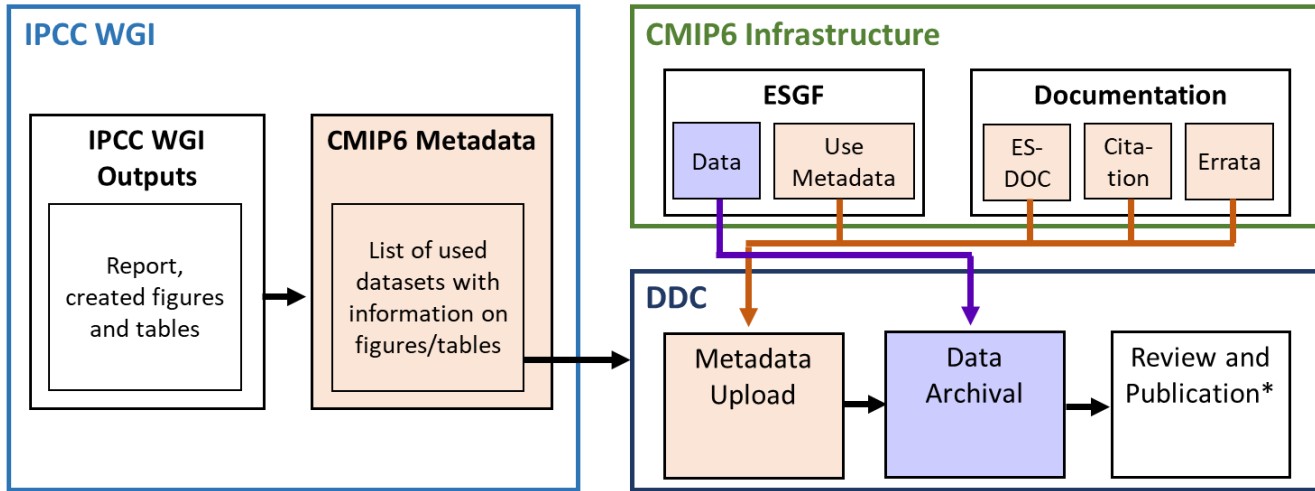

**Figure 6: CMIP6 input data archival workflow to build the DDC AR6 Reference Data Archive**

## 4    Changed role of the DDC Partner DKRZ in AR6

The implementation of the FAIR Guidelines expanded the role of the DDC Partner DKRZ from a responsibility limited to a
long-term data archive, operating mostly independent of WGI and the assessment cycle, to a more active partner with an enhanced role within the Sixth Assessment cycle. Close cooperation was required with the WGI TSU to formulate and implement the FAIR Guidelines. Thus, DDC Managers participated in the IPCC Expert Meeting on Assessing Climate Information for Regions in May 2018 and jointly organized the WGI Training on Data and Software Development in June 2019 together with the WGI TSU. Advice based on the DDC's long experience in data management was provided for gathering
the necessary information on data usage required of the authors, best practices in data citation and the definition of machine-actionable interfaces. The DDC Manager at DKRZ joined the WGI AR6 authors as contributing author and reviewed the First and Second Order Drafts of the report to provide expert advice on data management aspects.

This active role of the DDC in AR6 increased the DDC's visibility and resulted in requests for further support of the IPCC author teams during the preparation of the AR6. The DDC Partner DKRZ and former DDC Partner CEDA provided Virtual
Workspaces (Stockhause, 2020; Fig. 7) for the authors co-funded by the EU project Infrastructure for the European Network for the Earth System Modelling (IS-ENES, http://is.enes.org, last access: 29 June 2022). These collaboration platforms provided storage and compute resources for the chapter author groups together with access to requested core datasets and common software packages. Moreover, DKRZ supported the technical aspects of the ESMValTool (https://www.esmvaltool.org/, last access: 29 June 2022; Eyring et al., 2020) development and hosts the webpage with CMIP

evaluation results. On the national level, the DDC Manager at DKRZ joined the authors' subgroup of the German IPCC Coordination Office (https://www.de-ipcc.de/, last access: 29 June 2022) as German contributor to the IPCC AR6.

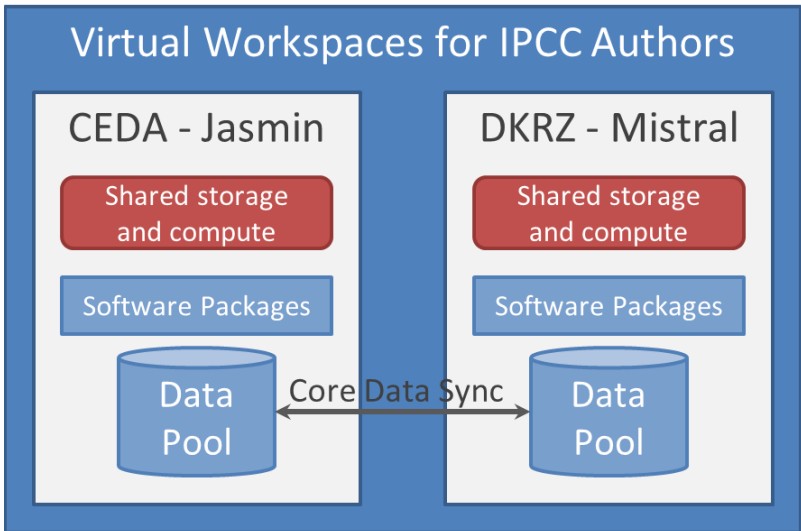

**Figure 7: Virtual Workspaces provided by CEDA and DKRZ for IPCC AR6 authors (co-funded by IS-ENES).**

During implementation of the FAIR Guidelines, questions arose that had to be solved with the IPCC Bureau. One of these involved the original licenses modelling groups attached to their CMIP6 data, which were too restrictive for a general reuse of IPCC data products, e.g. final data or Atlas data.  The IPCC had to ask the CMIP6 participants through the Working Group on Coupled Modelling (WGCM) for an exemption of the CMIP6 data licenses. As representatives of TG-Data, DDC Partner DKRZ and former DDC Partner CEDA were responsible for helping to ensure that IPCC technical requirements were met by

the CMIP infrastructure being developed under the coordination of the WGCM Infrastructure Panel (WIP, https://www.wcrp-climate.org/wgcm-cmip/wip, last access: 29 June 2022) and contributed data aspects to the IPCC Informal Group on Publications.

Independent of the FAIR Guidelines, the DDC Partners intensified their collaboration. The new UK DDC Partner MetadataWorks set up a joint DDC catalogue to improve the discovery of DDC data holdings. The DKRZ's DDC Manager

contributed to the development of the DDC's profile of the Data Catalog Vocabulary standard (W3C DCAT, https://www.w3.org/TR/vocab-dcat-3/, last access: 29 June 2022) and provided the metadata of its Reference Data Archive in December 2021. A central DDC help desk was set up to coordinate the DDC user support. The revision of the DDC webpages is ongoing with the aim of retiring outdated pages and refocusing the content on IPCC-related data, as called for under the renewed DDC Guidance.

**5        Position of the DDC within the climate infrastructure and role of CMIP6 for the AR6 cycle**

All of the IPCC Assessments have heavily drawn on the latest climate change research provided by the WCRP CMIP project. The core work of IPCC authors is the assessment of the latest peer-reviewed literature. CMIP data were used in the peer-reviewed literature and more directly for the creation of several IPCC report figures. With the introduction of the IPCC FAIR Guidelines, the dependency on CMIP-related literature and CMIP data were complemented by the dependency on CMIP6

infrastructure components (Petrie et al., 2021) and further DDC support activities (see section 4). For CMIP6, the WIP was formed by WGCM in 2014 to coordinate the development of the CMIP infrastructure across multiple institutions and agencies. The standardization of CMIP6 data is important for the reusability of the data. This includes compliance to the NetCDF/CF standard and specific file name conventions, a uniform directory structure, and the collection and dissemination of the CMIP6

Controlled Vocabularies (CMIP6-CVs, https://github.com/WCRP-CMIP/CMIP6_CVs, last access: 29 June 2022; Taylor et al., 2018).

The necessary infrastructure components include the data infrastructure Earth System Grid Federation (ESGF; Williams et al., 2016; Cinquini et al., 2014), which disseminates the data and provides use metadata and references to further information like data citation through its index. The CMIP6 Citation Service (http://cmip6cite.wdc-climate.de, last access: 29 June 2022; Stockhause and Lautenschlager, 2017) contributes data references and the discovery metadata gathered in the CMIP6-CVs to the long-term data archival (Stockhause et al., 2015) and thus links the data infrastructure to the long-term data preservation of the CMIP6 data subset in the DDC AR6 Reference Data Archive.

Apart from these necessary infrastructure components, ES-DOC provides detailed information on models, experiments and errata for further metadata enrichment in the DDC Reference Data Archive. The Citation Service succeeded in having full data citation coverage for all datasets published in the ESGF on the literature and data cut-off date for WGI AR6 on 31 January 2021, whereas the coverages of ES-DOC model descriptions and errata information are low. That means that the Citation Service and DDC mostly rely on the brief model descriptions in the CMIP6-CV provided by the CMIP6 participants during the registration process.

The agreed data standards and the high volume of the contributed data require thorough quality checks by the participants to ensure compliance and quality of the CMIP6 data. The conformance with the NetCDF/CF and the additional project metadata rules is automatically checked during ESGF data publication. The DDC complements them by metadata compliance and consistency checks.

At the same time, the DDC contributes unique services for the international climate community:

1. The DDC is the only data provider with a long-term commitment for data preservation and data services; it ensures that data will remain FAIR over time. Neither the ESGF data nodes nor the Copernicus Climate Data Store (CDS, https://cds.climate.copernicus.eu/, last access: 29 June 2022) have made such a commitment. Their focus lies on serving recent climate data.

2. The DDC preserves the data underpinning key statements of the Assessment Reports and thus the data on which several political decisions are based. The CMIP data subset in the DDC contains the scientific information and therefore is essential to trace back these decisions to the scientific basis.

3. As part of the IPCC Assessment process, the DDC reference climate data is quality-assured, enriched with metadata and made citable for its reusability by a variety of current and future applications.

4. The DDC supports the IPCC authors and the IPCC TSUs during the Assessment cycles.

DDC's data services need to be integrated not only in the IPCC AR6 products but also in the landscape of climate data infrastructures. Examples are the data provision through well-established domain data catalogs like the ESGF and through cross-domain infrastructures like the DataCite, the European Open Science Cloud (EOSC, https://eosc.eu/, last access: 29 June 2022) and the Nationale Forschungsdateninfrastruktur (NFDI, https://www.nfdi.de/, last access: 29 June 2022). Technical integration requires the exchange of standardized metadata and the implementation of standard interfaces, which are developed by the international organizations including W3C, ISO, Research Data Alliance (RDA), WDS, CODATA, Open Geospatial Consortium (OGC) or the Coalition for Publishing Data in the Earth and Space Sciences (COPDESS). The FAIR Digital Object Framework concept (De Smedt et al., 2019) provides guidance for the future interoperability of data and other digital objects. However, the most important collaboration partners for the DDC Partner DKRZ are CMIP, the WIP, ESGF, IS-ENES and further CMIP infrastructure partners. Through these collaborations, the DDC contributes its experiences to the CMIP future design.

## 6        Conclusion and Perspectives

The IPCC DDC has provided quality-assured, citable IPCC-relevant reference climate data for all IPCC Assessment Reports and has supported the IPCC Assessments over the 25 years of its existence. The specific role of the DDC has changed in order to adjust to evolving data management standards and evolving requirements from IPCC WGs. Furthermore, the responsibilities of the DDC Partner DKRZ have been adapted to developments in the CMIP6 infrastructure, which provides the data and documentation for the DDC's Reference Data Archive. DDC's data holdings provide valuable ancillary information for IPCC

Assessment Reports. AR6 marked a major change: The role of the DDC turned from maintaining an independent long-term data archive into providing general data services for the IPCC. At the same time, adoption of the IPCC FAIR Guidelines significantly enhanced the transparency of AR6 key findings. Their implementation posed a challenge to all partners: WGI TSU, IPCC authors and the DDC Partners. Data usage documentation in AR6 and long-term archival of related input and final datasets enable the traceability of results and the reuse of datasets. Long-term preservation of the data in the DDC ensures data

availability and traceability on the long-term. Still, the DDC AR6 data archive remains incomplete esp. in the long-term preservation of input datasets. This gap was identified by the DDC Partners in 2020 as one of several areas for future improvements:

1.        Exhaustive IPCC data archival,

2.        Improved global data access (e.g. compute service for reduction of data transfer volume to support DDC users in
developing countries and support for users from various domains),

3.        Data Discovery,

4.        Machine-accessible DDC data,

5.        Regional to local data and data services,

6.        Sustaining DDC Partners,

7.        Collaboration with data infrastructure networks, e.g. RDA, WDS or CODATA,

8.        Collaboration with cognate data providers, e.g. IPBES.

Some of these gaps have been filled with the limited DDC resources, like the collaboration aspects (gaps 7 and 8) or the data discovery issue (gap 3) with the establishment of the joint DDC catalogue, but the remaining gaps require funding and are related to the missing long-term strategy for the DDC (gap 6).

The current DDC Partner funding is provided by their IPCC member states for each Assessment cycle. In Germany, the DDC was funded as part of research projects supporting the German contribution to CMIP and enabled the DDC to add the Reference Data Archive for the each AR cycle. DDC operations and management are an in-kind contribution from DKRZ. Thus, long-term data preservation and maintenance of the DDC data services rely on voluntary contributions from institutions and individuals. National and institutional funders of the DDC as an international service expect other nations to share in the costs.

A new joint international funding approach for core data services and infrastructure components of the DDC is required. TG-Data has targeted DDC's sustainability as part of its AR6 review process with the aim to formulate recommendations for the AR7 data management. IPCC-internal options for DDC funding are:

1.    IPCC members fund DDC Partners for an Assessment cycle,

2.    IPCC members contribute to a DDC fund, or

3.    IPCC members funding WG TSUs also fund the associated DDC Partner.

The DDC Partner funding for an Assessment cycle (1) is problematic, since data from the departing DDC Partner are to be transferred to the replacing DDC Partner and the experience of the DDC Partner in IPCC processes and procedures is cyclically lost. The data volume of the Reference Data Archives of the DDC Partner DKRZ is high (see Fig. 3) and the transfer is time-consuming and expensive without adding any value. Furthermore, the important collaboration with CMIP6 and the various

infrastructure partners must be re-established by the new DDC Partner. Optimistically, this option will cause further significant but avoidable costs and, pessimistically, it is or will become impractical. Both option 2 and 3 can fund the addition of data for

a new Assessment cycle to the DDC Reference Data Archive, but only option 2 can additionally support the long-term aspects of data preservation and the provision of customized data services. Option 3 could pose a problem for the IPCC as it further increases the already high costs associated with a TSU and might discourage IPCC members from nominating a co-chair for a

WG. Option 2 requires Panel involvement and therefore may not yet be available for AR7.

External funding resources are restricted to public funders to protect IPCC's integrity. There are few international funders like the Belmont Forum. International organizations are in a similar situation to the IPCC and can offer letters of support, but rarely financial support. For example, WMO expressed its support for CMIP and the IPCC and emphasized the importance of data and infrastructure in a press release (WMO, 2019). Regarding the IPCC DDC, the German Minister of Education and Research,

Mrs. Stark-Watzinger, as representative of the German government explicitly mentioned Germany's involvement in the IPCC DDC and the importance of data for the IPCC process at the opening of the 55[th] Session of the IPCC and 12[th] Session of WGII on 10 February 2022 (IPCC, 2022). The value of data as scientific asset and the importance of open data has been recognized by several international organizations. In "The Beijing Declaration on Research Data", CODATA et al. (2019) emphasize9 the importance of the broad reuse of data to address global challenges and recognizes the enormous challenge in the

interoperability of data and responsible stewardship. The UNESCO states in its 'Recommendations on Open Science' (2021) that non-commercial infrastructures should facilitate ensuring the long-term preservation, stewardship and community control of research products including data. It recommends supporting these open infrastructures by direct funding and through an earmarked percentage of each funded grant.

This increased awareness of the importance of data as valuable scientific assets that need to be preserved and served to various

stakeholders over the long term facilitates the discussion on sustainable funding for the DDC.

**Appendix A – Core Variables of the Reference Data Archive**

```
air_pressure_at_sea_level
air_temperature
convective_precipitation_flux
dew_point_temperature
geopotential_height
global_average_thermosteric_sea_level_change
land_area_fraction
land_ice_area_fraction
large_scale_precipitation_flux
precipitation_flux
relative_humidity
sea_ice_amount
snowfall_amount
soil_moisture_content
specific_humidity
surface_altitude
surface_net_downward_shortwave_flux
surface_sensible_heat_flux
surface_downwelling_shortwave_flux_in_air
surface_snow_area_fraction
surface_snow_melt_flux
surface_snow_thickness
surface_temperature
water_evaporation_rate
wind_speed
x_wind
y_wind
```

**Table A1:** Core variables of the Reference Data Archive in CF Standard Name convention

**Code/Data availability**

No code nor data was created for this study.

**Author contribution**

MS wrote the manuscript draft and ML reviewed and contributed to the manuscript.

**Competing interests**

The authors declare that they have no conflict of interest.

**Acknowledgements**

We thank Tim Carter (former co-chair of IPCC TGICA, Finnish Environment Institute SYKE), who generously answered questions and shared unpublished materials about the early days of the IPCC DDC and TGCIA; Anna Pirani (head of IPCC WGI TSU) for the coordination of the formulation and FAIR Guidelines and their implementation in the AR6 WGI; and the DDC Partners, the WG TSUs esp. the WGI TSU colleagues, and the TG-Data members for their contributions to the IPCC FAIR Guidelines. Special thanks to the reviewers Karl Taylor, Paul Durack, and David Huard for their rich comments and 350 valuable suggestions for improving the manuscript.

The DDC Partner DKRZ has been funded by the Bundesministerium für Bildung und Forschung (BMBF) through several research grants. IS-ENES and the U.S. Department of Energy (DOE) have supported the ESGF and further CMIP services. The IS-ENES3 project has received funding from the European Union's Horizon 2020 research and innovation programme under grant agreement No 824084.

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
