# Peer review of "years IPCC Data Distribution Centre at DKRZ and the Reference Data Archive for CMIP data"

_Geoscientific Model Development, 2022_

## Author Response (AR1)

Answers to comments from Reviewers 1, 2 and 3 for:
"25 years IPCC Data Distribution Centre at DKRZ and the Reference Data Archive for CMIP data" by M. Stockhause and M. Lautenschlager

**General comments:**

Thanks to the reviewers for their detailed reviews with rich comments and valuable suggestions for improvements that add depth to the paper. The draft focused on the DDC and overemphasized technical aspects. The work of other groups providing standards and infrastructure were mentioned within the CMIP6 section but not for previous years. We added DDC services' dependencies on the efforts of these groups in the pre-CMIP6 sections to acknowledge their work. In this context, the position of the DDC within the complex project and infrastructure landscape was sharpened to answer the question of its relation to the CMIP data archive and the ESGF infrastructure. More details are given on the archival process and the long-term preservation and curation measures for the DDC's Reference Data Archive.

We took up the suggestion to discuss funding options and the long-term DDC strategy in the conclusion section, thereby shifting the focus away from technical towards institutional aspects. These changes are reflected in the abstract. The suggested figure on the DDC history has been introduced, enabling us to concentrate on the achievements/milestones in the text. The comment to expand the DDC history to provide reasons behind decisions was taken up exemplary for the EM on the future of TGICA. However, most of the IPCC-induced changes related to the Task Group rather than the DDC.

We have not addressed some comments that were outside the scope of this article, which is the balanced history of the DDC and the evolution of the role of the DDC Partner DKRZ. We do not discuss the contributions of our DDC Partners (e.g. the WGI Atlas) or further important IPCC-related input datasets outside of DKRZ's responsibility. We do not discuss CMIP and the CMIP infrastructure, but only their roles and importance for the DDC.

**Specific comments**:
- FAIR and IPCC FAIR Guidelines: The FAIR principles formulated by Wilkinson et al. (2016) were interpreted in multiple ways. For example, the GO-FAIR initiative focusses on the technical aspects and the role of PIDs. The IPCC's interpretation takes the essence of the FAIR principles and formulates a workflow approach to target them. IPCC also included the long-term aspect, which is not part of the FAIR principles but formulated in the TRUST principles for research repositories by Lin et al. (2020) and implemented in the CoreTrusSeal criteria, included in the WDS Regular Membership application process. In this regard, the IPCC FAIR Guidelines are more than an implementation of the FAIR principles but the specific data management concept of IPCC TG-Data aiming at enhanced transparency of the IPCC outputs and in principal allows the reproducibility of figures within the report.
- Target group of the IPCC FAIR Guidelines and TRUST principles/long-term data preservation: The IPCC FAIR Guidelines cover several data management aspects. We see the guidelines as collection of individual guidelines on specific aspects. The longterm data preservation aspect had been one part from the beginning (see Stockhause et al., 2019). It ensures reusability of the data on the long-term. It is a prerequisite for making data citable. Without data curation and keeping the data accessible, a data reference will not direct a reader to the data on the long-term. We do not think that the IPCC FAIR Guidelines formulated by Pirani et al. (2022) are specific enough to serve directly as IPCC authors but explain our approach towards FAIR data within the IPCC. IPCC authors need more specific and practical information on what to provide and in what form.

- Credit for "IPCC chapter scientists" for final data: Users citing the data give credit. A precondition is the long-term preservation and DOI assignment to the data. Thus, we added this credit aspect to the long-term preservation bullet point.

- CEDA's role: CEDA is no longer a formal DDC Partner, UK support has transitioned from CEDA to MetadataWorks, but CEDA is currently contributing to the AR6 final data archival.

- FAIR Guidelines' approval by TG-Data: We agree that the formulation was confusing and revised that part as suggested.

- Relation between downloads and citations: That is an important point but difficult to estimate. The data citation process is still broken in several aspects: 1. The cultural change to cite data in addition to papers is not completed. Many datasets are still not cited. 2. If the data is cited, many publishers do not expose data citations in their crossref metadata esp. for old papers. Thus as repository, we only know about a subset of cited datasets. In summary, the accessible data citations (using Scholix) are still too incomplete to give a reliable picture of the data citations.

- ES-DOC coverage: We rephrased the sentence to be neutral, but do not discuss the reasons for its poor coverage because of the paper's scope.

- Usage of Virtual Workspaces by IPCC authors: DKRZ did not separate IS-ENES or DKRZ users from IPCC authors. Some users might have multiple roles. We cannot give reliable numbers on IPCC authors using the Virtual Workspaces.

- Replica at other DDC Partners: The DDC Partners are responsible to ensure the long-term availability of their data shares, according to their answers to this CoreTrustSeal criterion. The DDC Partner DKRZ stores an off-cite copy of the DDC data at the Max Planck Computing and Data Facility (MPCDF) in Garching, Germany. We have included that in the text.

- Data requirements other than CMIP and CORDEX for the DDC: The role of the DDC is the long-term preservation of data underpinning the IPCC ARs. IPCC authors select the data and the WG TSUs gather this information on data usage.

- Downstream user requirements: Using model output data requires some knowledge on spatial-temporal resolution related to model grids and knowledge about experiment design and used standards. Outside of the modeling community, this knowledge is only partially present. Examples for downstream users are climate impact researchers or experts from insurance companies.

- Recognition of PCMDI's effort: We added that to the historical sections. This dependency on PCMDI's work for the pre-CMIP6 assessment cycles was important for the DDC and still is important as it lay the foundation for the current CMIP data standard.

- DDC gap analysis: These gaps were identified but not further analyzed due to higher priorities for the DDC tasks related to implementation of the IPCC FAIR Guidelines into the AR6 and the lack of staff to target these gaps.
- Chronological order in the history section: We want to introduce the DDC in the first part before we describe its history.
- Progress made in the IPCC AR6 WGI report and CMIP6 licenses was recognized.
- Suggested title "25-years of IPCC/CMIP data custodianship: the international Data Distribution Centre and the climate reference data archive": We decided not to change the title to keep the focus on the DDC Partner DKRZ and not move it towards CMIP.
- Figure 4 on the CMIP5 quality procedure: We have deleted it, as it was not clear and required more explanations than we have intended to provide in our review paper.
- Delete table 1: We moved the table to the Appendix, as it is not essential for the paper but provides further in-depths information on DDC's core variables.
- In-text URL references: To provide access dates for each URL is a journal's requirement. We discussed the reviewer's suggestions with Copernicus but Copernicus recommended to leave it as it is.
- Thanks for the additionally provided marked up word document.